# Correlations between Circulating and Tumor-Infiltrating CD4^+^ Treg Subsets with Immune Checkpoints in Colorectal Cancer Patients with Early and Advanced Stages

**DOI:** 10.3390/vaccines10091471

**Published:** 2022-09-05

**Authors:** Mohammad A. Al-Mterin, Khaled Murshed, Eyad Elkord

**Affiliations:** 1Natural and Medical Sciences Research Center, University of Nizwa, Nizwa 616, Oman; 2Department of Pathology, Hamad Medical Corporation, Doha 5207, Qatar; 3Biomedical Research Center, School of Science, Engineering and Environment, University of Salford, Manchester M5 4WT, UK

**Keywords:** correlation, FoxP3, Helios, Tregs, immune checkpoints, CRC

## Abstract

The existence of various T regulatory cell (Treg) subsets in colorectal cancer (CRC) could play a variety of functions in the regulation of anti-cancer immunity. We studied correlations between CD4^+^ Treg subsets with the expression of immunological checkpoints on CD4^+^ T cells, including PD-1, TIM-3, LAG-3, and CTLA-4 in CRC patients with early and advanced TNM staging. Strong positive correlations were found between frequencies of FoxP3^+^ Tregs and FoxP3^+^Helios^+^ Tregs with frequencies of various immune checkpoint-expressing CD4^+^ T cells in the tumor microenvironment (TME). However, there were strong negative correlations between frequencies of FoxP3^−^Helios^−^ T cells and these immune checkpoint-expressing CD4^+^ T cells. Specifically, in the TME, we found that the correlations between FoxP3^+^ Tregs, FoxP3^+^Helios^+^ Tregs, FoxP3^+^Helios^−^ Tregs, and FoxP3^−^Helios^−^ T cells with CD4^+^LAG-3^+^ T cells and CD4^+^CTLA-4^+^ T cells were higher in patients with early stages, suggesting the potential of these highly immunosuppressive cells in inhibiting inflammatory responses in the TME. However, the correlations between FoxP3^+^ Tregs, FoxP3^+^Helios^+^ Tregs, and FoxP3^−^Helios^−^ T cells with CD4^+^TIM-3^+^ T cells were higher in patients with advanced stages. This is the first study to explore correlations of Treg subpopulations with immune checkpoint-expressing CD4^+^ T cells in CRC based on clinicopathological features of CRC patients. The findings of our study provide a justification for focusing on these cells that possess highly immunosuppressive features. Understanding the correlations between different immune checkpoints and Treg subsets in CRC patients has the potential to enhance our understanding of core mechanisms of Treg-mediated immunosuppression in cancer.

## 1. Introduction

Colorectal cancer (CRC) is one of the malignancies with the highest incidence worldwide [1]. Chronic secretion of inflammatory cytokines by intestinal cells could lead to the development of cancer [2]. Tumor staging is one of the most significant prognostic factor in CRC [3], which has been affected by the extent of intestinal invasion, lymph node metastasis, and distant metastases [3]. Immune checkpoints (ICs) and their ligands are typically seen at increased levels in the tumor microenvironment (TME) of different malignancies, which result in the inhibition of anti-tumor immunity [4,5]. Specifically, one of the resistance mechanisms of tumor cells against immunological responses is by modulating the expression of certain ICs [6,7]. T regulatory cells (Tregs) are immunosuppressive cells that play an important role in the maintenance of immune system homeostasis, self-tolerance, and regulation of cancer immunity [8]. Tregs are a substantial subset of CD4^+^ T cells that express higher levels of the interleukin-2 receptor alpha chain (CD25) and the transcription factor FoxP3 [9]. The presence of high levels of infiltrated Tregs in malignancies has been linked to poor clinical outcomes in different types of cancer [10]. However, in CRC, the function of Tregs remains controversial. Several studies have shown that high levels of infiltrated FoxP3^+^ Tregs in tumor tissues are associated with a better prognosis in CRC patients [7,11,12,13]. Helios is a key transcription factor that is essential for FoxP3^+^ Treg inhibitory action and functional stability [14,15,16,17]. Interestingly, a high level of Helios mRNA was found in tumor tissues in CRC advanced stages, signifying their potential effects in CRC progression [18]. We have recently reported that some ICs, including T cell immunoglobulin and mucin domain-containing protein-3 (TIM-3), cytotoxic T lymphocyte-associated antigen-4 (CTLA-4), programmed cell death-1 (PD-1), and lymphocyte-activation gene-3 (LAG-3), were upregulated on CD4^+^ T cells, and they play roles in CRC disease progression [7]. In this study, we investigated any potential correlations between frequencies of different Treg subsets with CD4^+^ T cell expressing different immune checkpoints according to the tumor staging.

## 2. Materials and Methods

### 2.1. Patients and Samples

Correlation analyses were performed on a total of 32 colorectal cancer (CRC) patients who fulfilled the requirements to be included in the study. Tumor and normal colon samples were collected from 22 out of the 32 patients. Hamad Medical Corporation in Doha, Qatar, provided the required ethical approval to conduct this research, which was carried out in accordance with their guidelines (Protocol no. MRC-02-18-012). None of the patients have had any kind of therapy prior to surgeries, and they all provided their written informed consent before samples were collected. Pathological features of patients who participated in this research are presented in Al-Mterin et al. [19].

### 2.2. Multi-Parametric Flow Cytometry

In this research, there were no more experiments performed. Immune staining and flow cytometry analyses have been performed in accordance with our previously published paper [7]. Briefly, peripheral blood mononuclear cells (PBMCs) were washed and suspended in flow cytometry staining buffer. Fc receptors (FcR) were first blocked using FcR Blocker (Miltenyi Biotec). Samples were then stained using particular antibodies that recognized cell surface or intracellular antigens. Samples were run on a BD LSRFortessa X-20 SORP flow cytometer, and data were collected using BD FACSDiva software (BD Biosciences), and then analyzed by FlowJo V10 software (FlowJo, Ashland, OR, USA).

### 2.3. Statistical Analyses

GraphPad Prism 9 program was used in order to perform the correlation investigations (GraphPad Software, California, CA, USA). In order to determine whether or not the datasets were normally distributed, the Shapiro–Wilk test was performed. Normally distributed data were analyzed using Pearson’s correlation test, while Spearman’s rank correlation test was used for samples that did not show normal distribution. Correlation analyses were performed between early (I and II) and advanced (III and IV) CRC stages. *p* value of ≤0.05 was considered to be statistically significant.

## 3. Results

### Correlations between Frequencies of Treg Subsets and IC-Expressing CD4^+^ T Cells Based on CRC Stages

We have previously determined frequencies of different CD4^+^ T cell subsets in peripheral blood mononuclear cells (PBMCs), normal tissue-infiltrating lymphocytes (NILs), and tumor-infiltrating lymphocytes (TILs) of CRC patients [7]. In addition, we investigated the correlations between frequencies of FoxP3^+^ Tregs and FoxP3^±^Helios^±^ T cells with frequencies of ICs expressed on CD4^+^ T cells in CRC patients [19]. In this study, we further identified the correlations between frequencies of FoxP3^+^ Tregs with Helios^+^ T cells and IC-expressing CD4^+^ T cells in PBMCs, NILs, and TILs in CRC patients based on CRC stages. Correlations between frequencies of FoxP3^+^ Tregs and Helios^+^ T cells in circulation were stronger in advanced tumor stages, compared to early stages (correlation coefficient r = 0.582, *p* = 0.036 [early]; r = 0.880, *p* < 0.0001 [advanced]) (Figure 1A). Conversely, such correlations were stronger in early stages, compared to advanced stages in the TME (r = 0.955, *p* < 0.0001 [early]; r = 0.635, *p* = 0.019 [advanced]) (Figure 1A). Interestingly, these correlations were not observed in normal colon tissues (Figure 1A), indicating their potential roles in the TME. It is speculated that highly suppressive FoxP3^+^Helios^+^ Tregs could play harmful roles in inhibiting anti-tumor immune responses in the circulation, especially at CRC advanced stages, while these cells could be beneficial in controlling inflammation in the TME especially at CRC early stages.

We then investigated the correlations between CD4^+^ Treg subsets with CD4^+^ T cells expressing different ICs, based on tumor stages (Figure 1B–E). Correlations between frequencies of FoxP3^+^ Tregs and CD4^+^PD-1^+^ T cells in TME were stronger in early tumor stages, compared to advanced stages (r = 0.675, *p* = 0.045 [early]; r = 0.489, *p* = 0.089 [advanced]) (Figure 1B). Such correlations were stronger in early stages, compared to advanced stages in the NILs (r = r = 0.721, *p* = 0.023 [early]; r = 0.516, *p* = 0.085 [advanced]) (Figure 1B). Remarkably, these correlations were not observed in circulation (Figure 1B). Correlations between frequencies of FoxP3^+^ Tregs and CD4^+^TIM-3^+^ T cells in TME were stronger in advanced tumor stages, compared to early stages (r = 0.737, *p* = 0.023 [early]; r = 0.818, *p* = 0.0006 [advanced]) (Figure 1C). However, these correlations were not observed in circulation and normal colon tissues (Figure 1C), indicating their potential roles in the TME in advanced tumor stages. A moderate correlation was observed between frequencies of FoxP3^+^ Tregs and CD4^+^LAG-3^+^ T cells in PBMCs of CRC patients, and it was found to be stronger in advanced tumor stages compared to early stages (r = −0.022, *p* = 0.945 [early]; r = 0.488, *p* = 0.039 [advanced]). The correlation in TILs was stronger in early tumor stages, compared to advanced stages (r = 0.833, *p* = 0.008 [early]; r = 0.685, *p* = 0.012 [advanced]) (Figure 1D). Interestingly, correlations between frequencies of FoxP3^+^ Tregs and CD4^+^CTLA-4^+^ T cells were stronger in early stages, compared to advanced stages in the TME (r = 0.930, *p* = 0.0003 [early]; r = 0.637, *p* = 0.034 [advanced]). These correlations were not observed in PBMCs or NILs (Figure 1E).

We went one step further and determined correlations between the various FoxP3^+^Helios^+/−^ T cell subsets and the distinct immunological checkpoints expressed on CD4^+^ T cells. Correlations between frequencies of FoxP3^+^Helios^+^ Tregs and CD4^+^PD-1^+^ T cells were stronger in early stages, compared to advanced stages in the TME, but not significant (r = 0.644, *p* = 0.061 [early]; r = 0.322, *p* = 0.282 [advanced]) (Figure 2A). Furthermore, a strong positive correlation was observed between frequencies of FoxP3^+^Helios^+^ T cells and CD4^+^PD-1^+^ T cells in NILs in CRC patients, and it was found to be stronger in early tumor stages compared to advanced stages (r = 0.818, *p* = 0.005 [early]; r = 0.393, *p* = 0.206 [advanced]) (Figure 2A). However, these correlations were not observed in normal colon tissues. Correlations between frequencies of FoxP3^+^Helios^+^ Tregs and CD4^+^TIM-3^+^ T cells in TME were stronger in advanced tumor stages, compared to early stages (r = 0.793, *p* = 0.011 [early]; r = 0.771, *p* = 0.002 [advanced]) (Figure 2B). However, these correlations were not observed in circulation and normal colon tissues (Figure 2B). A moderate correlation between frequencies of FoxP3^+^Helios^+^ Tregs and CD4^+^LAG-3^+^ T cells in in circulation was stronger in advanced tumor stages, compared to early stages (r = −0.146, *p* = 0.631 [early]; r = 0.478, *p* = 0.044 [advanced]) (Figure 2C). However, such correlation was stronger in early stages, compared to advanced stages in TME (r = 0.833, *p* = 0.008 [early]; r = 0.602, *p* = 0.032 [advanced]), and in normal colon tissues (r = 0.757, *p* = 0.015 [early]; r = −0.037, *p* = 0.908 [advanced]) (Figure 2C). Strong correlations were seen between frequencies of FoxP3^+^Helios^+^ Tregs and CD4^+^CTLA-4^+^ T cells in PBMCs in patients with advanced tumor stages (r = 0.162, *p* = 0.594 [early]; r = 0.550, *p* = 0.018 [advanced]), while such correlation was observed in the TME of patients with early tumor stage (r = 0.911, *p* = 0.0006 [early]; r = 0.486, *p* = 0.129 [advanced]) (Figure 2D). However, these correlations were not observed in normal colon tissues (Figure 2D).

With regards to the less suppressive/stable FoxP3^+^Helios^−^ Tregs, a moderate correlation was found between frequencies of these Tregs and CD4^+^PD-1^+^ T cells in PBMCs of patients with advanced tumor stages (r = −0.188, *p* = 0.557 [early]; r = 0.499, *p* = 0.035 [advanced]), and a strong correlation in the TME of patients with advanced tumor stages (r = 0.573, *p* = 0.106 [early]; r = 0.635, *p* = 0.019 [advanced]), and in NILs in patients with early stages (r = 0.660, *p* = 0.043 [early]; r = 0.370, *p* = 0.235 [advanced]) (Figure 3A). Unlike FoxP3^+^Helios^+^ Tregs, there were no correlations between the frequencies of FoxP3^+^Helios^−^ Tregs and CD4^+^TIM-3^+^ in PBMCs, TILs, and NILs (Figure 3B). Strong correlations were found between frequencies of FoxP3^+^Helios^−^ Tregs with CD4^+^LAG-3^+^ T cells and CD4^+^CTLA-4^+^ T cells in TILs in patients with early tumor stages, compared to advanced stages (r = 0.800, *p* = 0.013 [early]; r = 0.214, *p* = 0.478 [advanced]; r = 0.824, *p* = 0.006 [early]; r = 0.529, *p* = 0.093 [advanced], respectively) (Figure 3C,D). Also, a strong correlation was found between frequencies of FoxP3^+^Helios^−^ Tregs with CD4^+^CTLA-4^+^ T cells in NILs in patients with advanced stages (r = 0.550, *p* = 0.132 [early]; r = 0.828, *p* = 0.005 [advanced] (Figure 3D).

Conversely, negative correlations were found between frequencies of FoxP3^−^Helios^−^ T cells and CD4^+^PD-1^+^ T cells in the TME in patients with early and advanced tumor stages (r = −0.662, *p* = 0.052 [early]; r = −0.576, *p* = 0.039 [advanced]) (Figure 4A), and in NILs the correlation was stronger in early tumor stages, compared to advanced stages (r = −0.709, *p* = 0.026 [early]; r = −0.430, *p* = 0.162 [advanced]) (Figure 4A). Moreover, strong negative correlations were found between frequencies of FoxP3-Helios- non-Tregs and CD4^+^TIM-3^+^ T cells in TILs, and they were found to be stronger in advanced tumor stages compared to early stages (r = −0.751, *p* = 0.019 [early]; r = −0.812, *p* = 0.0007 [advanced]) (Figure 4B). Furthermore, strong negative correlations were found between frequencies of FoxP3^−^Helios^−^ T cells and CD4^+^LAG-3^+^ T cells in TILs in patients with early tumor stages (r = −0.833, *p* = 0.008 [early]; r = −0.506, *p* = 0.079 [advanced]) (Figure 4C). The frequencies of FoxP3^−^Helios^−^ T cells and CD4^+^CTLA-4^+^ T cells in PBMCs were found to have stronger negative correlations in advanced tumor stages, compared to early stages (r = −0.041, *p* = 0.903 [early]; r = −0.651, *p* = 0.003 [advanced]). Conversely, such correlations were stronger in early stages, compared to advanced stages in the TME (r = −0.892, *p* = 0.002 [early]; r = −0.702, *p* = 0.018 [advanced]) (Figure 4D). The correlation in NILs was stronger in advanced tumor stages, compared to early stages (r = −0.316, *p* = 0.410 [early]; r = −0.925, *p* = 0.0003 [advanced]) (Figure 1D). Table 1 summarizes the correlations between different CD4^+^ Treg subsets and ICs-expressing CD4^+^ T cells in early and advanced stages.

## 4. Discussion

CRC treatment choices and prognosis are dependent on the histopathological factors that have a direct association with survival and tumor relapse. Understanding the underlying causes of CRC is critical for cancer prevention and therapy [20].

Most of CD4^+^ T cells in CRC are mainly T regulatory cells, expressing various IC molecules such as TIM-3, LAG-3, CTLA-4, and others [7]. High expressions of Treg-related markers were found in the TME in CRC, indicating their potential roles in carcinogenesis [7,21,22]. CRC patients with high levels of tumor-infiltrating FoxP3^+^ Tregs had a higher chance of survival, in contrast to those with other types of solid tumors [13,23]. Moreover, a high frequency of FoxP3^+^ Tregs in the TME lead to a favorable outcome in CRC patients, indicating that the presence of FoxP3^+^ Tregs is one of the most helpful predictors of disease prognosis in CRC patients [12,24]. On the other side, it has been shown that circulating Tregs are efficient in inhibiting antitumor immunity, which results in a negative outcome in patients with CRC [25,26]. Colorectal tumor tissues showed higher frequency of Helios^+^ Tregs than PBMC and normal colon tissue [23,27], indicating that Helios may have a role in CRC progression [18]. We have recently reported a robust correlation between FoxP3^+^ and Helios^+^ expression in TILs and PBMCs in CRC patients [16]. In addition, we reported in a previous study strong positive correlations between frequencies of CD4^+^Helios^+^ T cells and FoxP3^+^ Tregs in TILs and PBMCs, confirming that intratumoral and circulating FoxP3^+^ Tregs have higher expression of Helios [19], indicating their immunosuppressive features and highly activated status [5,28]. In line with these findings, we observed a significant positive correlation between the frequencies of FoxP3^+^ Tregs and CD4^+^Helios^+^ T cells in the TME, which was stronger in early stages, compared to advanced stages, suggesting the potentials of these highly immunosuppressive cells in inhibiting inflammatory responses in the TME. However, these correlations were stronger in advanced stages, compared to early stages in circulation, suggesting the potential of this Treg subset in inhibiting anti-tumor immune responses in circulation. could be used as predictors of disease prognosis in CRC patients.

The suppressive effect of Tregs varies based on their expression of inhibitory IC receptors such as TIM-3, PD-1, LAG-3, and CTLA-4 [29,30,31,32]. In addition, elevated numbers of infiltrating Tregs expressing ICs inhibit the activation of CD8^+^ and CD4^+^ T cells within the tumor [33]. Recently, our group found that key ICs, including CTLA-4, TIM-3, LAG-3, and PD-1, were highly expressed on CD4^+^ T cells in the TME of CRC [7]. We also found different associations between Treg subsets expressing ICs and disease-free survival (DFS) in CRC patients [34]. Specifically, high frequencies of CD4^+^ T cell subsets expressing PD-1 and CTLA-4 were associated with worse DFS in circulation. However, high frequencies of Tregs/CD4^+^ T cells expressing TIM-3 in circulation were associated with longer DFS [34]. Additionally, our group found PD-1 mRNA level in CRC tumor tissues was shown to be considerably higher in CRC patients with early stages [18]. In our previous study, we found strong correlations between frequencies of CD4^+^PD-1^+^ T cells with different subsets of Tregs/T cells including FoxP3^+^ Tregs, FoxP3^+^Helios^+^ Tregs, and FoxP3^+^Helios^−^ Tregs in NILs and TILs [19]. In line with these findings, we found strong positive correlations between frequencies of FoxP3^+^ Tregs and FoxP3^+^Helios^+^ Tregs with CD4^+^PD-1^+^ T cells in TILs and NILs in early tumor stages. Also, positive correlations were found between frequencies of FoxP3^+^Helios^−^ Tregs and CD4^+^PD-1^+^ T cells in PBMCs and TILs in advanced tumor stages, and a positive correlation in NILs only in early tumor stages. A strong negative correlation was observed between frequencies of FoxP3^−^Helios^−^ T cells and CD4^+^PD-1^+^ T cells in TILs in both early and advanced tumor stages, and in NILs with early tumor stages. These results indicate that PD-1 expression is not induced only by tumor cells. The expression of PD-1 in NILs might explain the side effects of anti-PD-1 antibodies [35]. In addition, categorizing CRC patients based on disease stages and identifying the exact CD4^+^ T cell subpopulations could help in targeting PD-1 in CRC patients in more specific ways, and reduce the side effects of current anti-PD1 antibodies.

A tumor immunosuppressive environment may be exacerbated by the overexpression of TIM-3 on CD4^+^ T cells [36]. Additionally, another study demonstrated that TIM-3 is highly expressed on both CD8^+^ and CD4^+^ T cells in TILs, but insignificantly in PBMCs in human lung cancer tissues [32]. Moreover, a recent study found that high levels of TIM-3 were expressed in CRC tissues, compared with normal tissues. This expression was significantly associated with tumor size, tumor-node-metastasis staging, and distant metastasis [37]. Furthermore, our group found that the TIM-3 mRNA level in CRC tissues was increased in advanced tumor stages, proposing their possible roles in CRC progression [18]. In a previous study, we found strong correlations between frequencies of CD4^+^TIM^+^ T cells and FoxP3^+^ Tregs, Helios^+^ T cells, FoxP3^−^Helios^−^ T cells, FoxP3^+^Helios^+^ Tregs, and FoxP3^+^Helios^−^ Tregs in TILs [19]. In addition, the expression of TIM-3 in the CRC tissues was significantly associated with distant metastasis, tumor-node-metastasis staging, and tumor size [37]. Consistent with these data, we found that correlations between frequencies of FoxP3^+^Helios^+^ Tregs and CD4^+^TIM-3^+^ T cells in TME were stronger in advanced tumor stages, suggesting that TIM-3 could be a critical mediator in the progression of CRC and may be considered a possible therapeutic target for the treatment of CRC. Moreover, such correlations could be used to distinguish early vs. advanced stages of CRC.

Tregs expressing LAG-3 are highly suppressive and proliferative in CRC patients [7,31]. In CRC patients with significant microsatellite instability, increased LAG-3 expression was associated with a poor prognosis [38]. A recent study showed that upregulation of LAG-3 on tumor tissues was associated with a bad prognosis in CRC patients with microsatellite instability-high [38]. Moreover, compared to the normal group, the gene expression of LAG-3 was shown to be considerably elevated in CRC patients [39]. In this study, we found strong correlations between Tregs expressing LAG-3^+^ in circulation of advanced CRC, indicating these cells may have a potent suppressor activity [31]. Interestingly, a study found that LAG-3 expression on TILs was significantly associated with better 5-year DFS in early stages of CRC [40]. Accordingly, we found strong correlations in early CRC stages in the TME, indicating the beneficial anti-inflammatory role of CD4^+^LAG-3^+^ T cells in the TME of CRC patients. In addition, this might assist to predict outcomes for individuals with early stages of colon cancer and perhaps identify those who could benefit from adjuvant chemotherapy.

A significant correlation has been reported between CTLA-4 and FoxP3 expression in PBMCs of patients with breast cancer [41]. Additionally, Toor et al. found that levels of CD4^+^CTLA-4^+^ T cells increased significantly in circulation of CRC patients with advanced stages, indicating a link between higher levels of CTLA-4^+^ Tregs and CRC progression [7]. Importantly, in advanced stages of colorectal cancer, the levels of CTLA-4 mRNA in tumor tissues were shown to be higher [18]. In our previous study, a significant correlation was observed between frequencies of CD4^+^CTLA-4^+^ T cells with different CD4^+^Treg/T cell subsets in CRC patients [19]. Consistent with these data, we found strong correlations between frequencies of CD4^+^CTLA-4^+^ T cells and FoxP3^+^ Tregs, FoxP3^+^Helios^+^ Tregs, and FoxP3^−^Helios^−^ T cells in the TME of CRC patients with early stages, indicating the beneficial immunosuppressive role of CD4^+^CTLA-4^+^ T cells in the TME of CRC patients. More immunosuppressive cells with high expressions of CTLA-4 are good in early stages in the TME. Moreover, we found correlations between frequencies of CD4^+^CTLA-4^+^ T cells and Tregs/T cells in NILs of advanced CRC patients, suggesting that CTLA-4 is not induced only by the TME. Indeed, the correlations between CD4^+^CTLA-4^+^ T cells and Treg subgroups in circulation of patients with advanced stages could lead to the inhibition of antitumor immunity, which results in a negative outcome in patients with CRC. Furthermore, expression of CTLA-4 in TILs in early stages could have an anti-inflammatory effect in the TME of CRC patients. Nonetheless, further investigations are needed to validate these findings in larger cohorts of patients. Further experiments need to be done in order to determine roles of these subsets in the functional regulation of Tregs in the tumor tissues and circulation of patients with CRC. Additionally, more investigations are needed to determine biomarkers that could be used to predict clinical outcomes of chemotherapy or surgery. A recent study found that patients with high PD-L1 expression in tumor tissues had significantly shorter DFS than patients with low expression. Moreover, high expression of PD-L1 on tumor-infiltrating mononuclear cells were associated with longer DFS in patients with stage III CRC who endured curative surgery and adjuvant chemotherapy [42]. Further research into the relationships between distinct Treg subgroups and T cells co-expressing various immunological checkpoints would be intriguing.

## 5. Conclusions

To our knowledge, this is the first study to examine the correlations between distinct Treg subsets and CD4^+^ T cells expressing ICs in CRC patients according to disease stages. In this study important findings demonstrated that frequencies of different Treg subsets in the TME correlate with frequencies of certain IC-expressing CD4^+^ T cells in CRC patients with different disease stages. Moreover, these correlations could be used as prognostic biomarkers for disease staging. Understanding the correlations between different ICs and Treg subsets in CRC patients with different clinicopathological statuses has the potential to enhance our understanding of the core mechanisms of Treg-mediated immunosuppression in cancer. The findings of our study provide a justification for focusing on these cells that possess highly immunosuppressive features.

## Figures and Tables

**Figure 1 vaccines-10-01471-f001:**
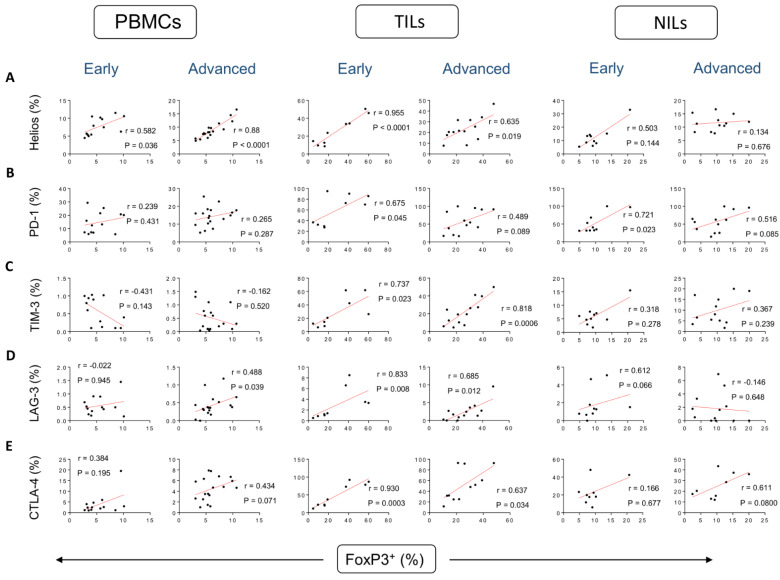
Correlations between frequencies of FoxP3^+^ and immune checkpoints in CD4^+^ T cells in CRC patients based on disease stages. Correlations between frequencies of CD4^+^FoxP3^+^ Tregs with Helios^+^ (**A**), PD-1^+^ (**B**), TIM-3^+^ (**C**), LAG-3^+^ (**D**), and CTLA-4^+^ (**E**) in PBMCs, TILs, and NILs in CRC patients with early and advanced stages.

**Figure 2 vaccines-10-01471-f002:**
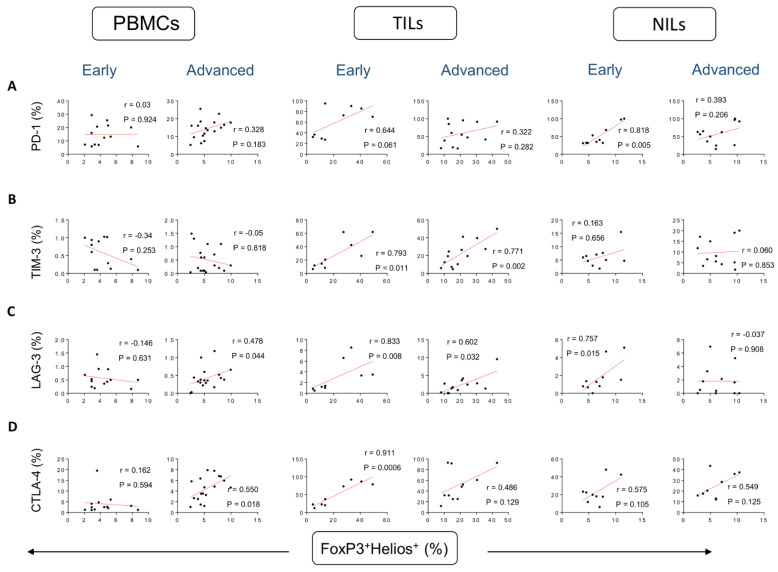
Correlations between frequencies of FoxP3^+^Helios^+^ and immune checkpoints in CD4^+^ T cells in CRC patients based on disease stages. Correlations between frequencies of CD4^+^FoxP3^+^Helios^+^Tregs with PD-1^+^ (**A**), TIM-3^+^ (**B**), LAG-3^+^ (**C**), and CTLA-4^+^ (**D**) in PBMCs, TILs, and NILs in CRC patients with early and advanced stages.

**Figure 3 vaccines-10-01471-f003:**
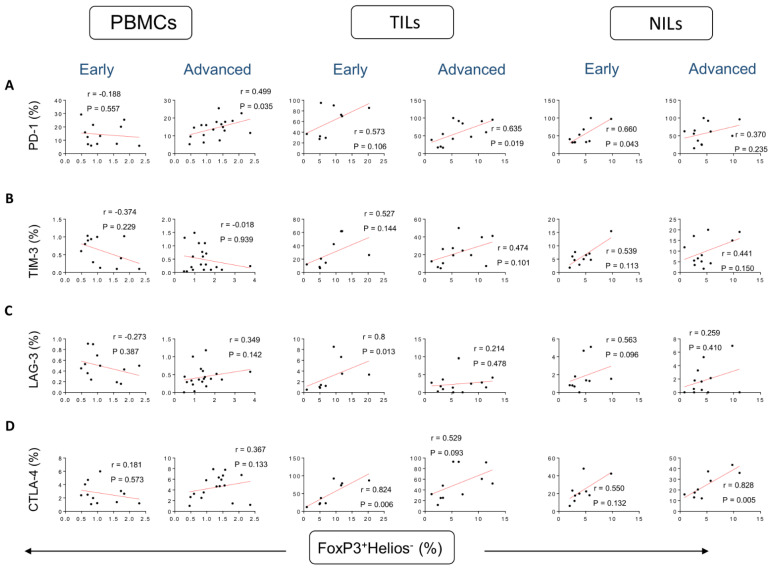
Correlations between frequencies of FoxP3^+^Helios^−^ and immune checkpoints in CD4^+^ T cells in CRC patients based on disease stages. Correlations between frequencies of CD4^+^FoxP3^+^Helios^−^Tregs with PD-1^+^ (**A**), TIM-3^+^ (**B**), LAG-3^+^ (**C**), and CTLA-4^+^ (**D**) in PBMCs, TILs, and NILs in CRC patients with early and advanced stages.

**Figure 4 vaccines-10-01471-f004:**
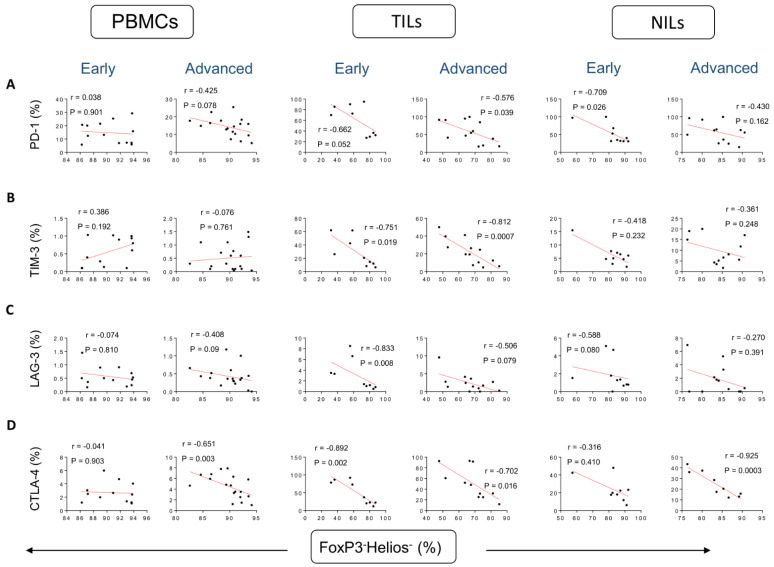
Correlations between frequencies of FoxP3^−^Helios^−^ and immune checkpoints in CD4^+^ T cells in CRC patients based on disease stages. Correlations between frequencies of CD4^+^FoxP3^−^Helios^−^Tregs with PD-1^+^ (**A**), TIM-3^+^ (**B**), LAG-3^+^ (**C**), and CTLA-4^+^ (**D**) in PBMCs, TILs, and NILs in CRC patients with early and advanced stages.

**Table 1 vaccines-10-01471-t001:** Summary of the significant correlations between different CD4^+^ Treg/T cell subsets and ICs expression in early and advanced stages.

	PD-1			TIM-3			LAG-3			CTLA-4		
	PBMCs	TILs	NILs	PBMCs	TILs	NILs	PBMCs	TILs	NILs	PBMCs	TILs	NILs
FoxP3^+^	NC	E	E	NC	E/A *	NC	A	E */A	NC	NC	E */A	NC
FoxP3^+^Helios^+^	NC	NC	E	NC	E/A *	NC	A	E */A	E	A	E	NC
FoxP3^+^Helios^−^	A	A	E	NC	NC	NC	NC	E	NC	NC	E	A
FoxP3^−^Helios^−^	NC	E/A *	E	NC	E/A *	NC	NC	E	NC	A	E */A	A

E: early stage, A: advanced stage, NC: no correlation, *: correlation is higher in this stage.

## Data Availability

The data presented in this study are available on request from the corresponding author.

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
