# Peer review of "Correlations between Circulating and Tumor-Infiltrating CD4+ Treg Subsets with Immune Checkpoints in Colorectal Cancer Patients with Early and Advanced Stages"

_vaccines, 2022, doi:10.3390/vaccines10091471_

Round 1

Reviewer 1 Report

The Authors examine the correlations between distinct Treg subsets and CD4+ T cells expressing ICs in CRC patients according to disease stages. The data are original and interesting. I did not find any shortcomings in the experimental design or in the discussion of the data, which is thorough and comprehensive.

The Authors investigate the correlations between frequencies of FoxP3+ Tregs with Helios+ T cells and IC-expressing CD4+ T cells in PBMCs, NILs and TILs  in CRC patients based on CRC stages. They find that correlations between frequencies of FoxP3+ Tregs and Helios+ T cells in circulation are stronger in advanced tumor stages compared to early stages, while in TME such correlations are stronger in early stages compared to advanced stages. They speculate that FoxP3+Helios+ Tregs play harmful roles in inhibiting anti-tumor immune responses in the circulation at CRC advanced stages, while could control inflammation in the TME at CRC early stages. They then investigate the correlations between CD4+ Treg subsets with CD4+ T cells expressing different ICs, based on tumor stages, and between the various FoxP3+Helios+/- T cell subsets and the distinct immunological checkpoints expressed on CD4+ T cells. As they expose in the detailed discussion, frequencies of different Treg subsets in the TME correlate with frequencies of certain IC-expressing CD4+ T cells in CRC patients with different disease stages. To the best of my knowledge, correlations between distinct Treg subsets and CD4+ T cells expressing ICs in CRC patients according to disease stages had not been investigated so far. Therefore this study appear to be innovative and the data reported by the authors provide relevant indications on the disease mechanism in the whole organism and on prognosis. The authors' documented experience in the technology used and in the study of immunity in tumors allows them to formulate reliable analyses and conclusions in the complex field of Treg populations study.

Author Response

We would like to thank the reviewer for all these positive comments on our work and this manuscript. We really appreciate highlighting the main findings and the importance of this study.

Reviewer 2 Report

In the manuscript submitted by Al-Mterin et al, the authors reported the finding that certain CD4+ Treg subsets were correlated with immune checkpoints in either early or advanced stage of colorectal cancer (CRC).  This is a further analysis of their previous work in which the correlation between certain CD4+ subsets with immune checkpoints in CRC patients. Although the authors provided a rather detailed analysis to demonstrate the correlation between them, the main concern is still the lack of clear clinical application of the findings.  For instance, is there a specific correlation could be used to distinguish early vs advance stage of CRC? Or to predict the outcome of surgery/chemotherapy? Or disease-free survival/relapse, etc.? And could the finding be verified in other patients?  These should be further discussed and stated in the discussion and conclusion section, other than just vaguely saying “to enhance our understanding …….”, which is basically nothing.

Author Response

In the manuscript submitted by Al-Mterin et al, the authors reported the finding that certain CD4+ Treg subsets were correlated with immune checkpoints in either early or advanced stage of colorectal cancer (CRC).  This is a further analysis of their previous work in which the correlation between certain CD4+ subsets with immune checkpoints in CRC patients. Although the authors provided a rather detailed analysis to demonstrate the correlation between them, the main concern is still the lack of clear clinical application of the findings.

Thank you for your comments. Finding such correlations are quite important. We appreciate that the reviewer can agree that the clear clinical applications of these correlations require further investigations and long-term follow up. We have revised the manuscript according to the reviewer’s comments. 

As reviewer 1 correctly mentioned in their comments that “correlations between distinct Treg subsets and CD4+ T cells expressing ICs in CRC patients according to disease stages had not been investigated so far. Therefore, this study appears to be innovative and the data reported by the authors provide relevant indications on the disease mechanism in the whole organism and on prognosis”.

For instance, is there a specific correlation could be used to distinguish early vs advance stage of CRC?

Thank you for your comment. We found that correlations between frequencies of different Treg subsets and CD4+TIM-3+ T cells in TME were stronger in advanced tumor stages, suggesting that TIM-3 could be used as a marker to distinguish early vs advanced stage of CRC. Based on the reviewer’s comment, we have mentioned this result in page 9, lines 281-282.

Or to predict the outcome of surgery/chemotherapy?

All patients included in this study were treatment-naïve prior to sample collection. In this cohort of patients, patients with early stages CRC had been treated with surgery, while patients with advanced stages CRC received chemotherapy after surgery. As both groups were treated with surgery, it is impossible to determine any biomarker to discriminate the outcomes. However, we have referred to a new study, where it has been reprted that PD‐L1 expression status can be a predictor of recurrence for patients with stage III colorectal cancer who underwent curative surgery and adjuvant chemotherapy (please see page 10, lines 318-323, ref 42).

Or disease-free survival/relapse, etc.? And could the finding be verified in other patients?  These should be further discussed and stated in the discussion and conclusion section, other than just vaguely saying “to enhance our understanding …….”, which is basically nothing

Thank you for your suggestions. We have already added in the discussion section about different associations between Treg subsets expressing ICs and disease-free survival (DFS), page 9, lines 245-250. In our previous work, we found different associations between Treg subsets expressing ICs and disease-free survival (DFS) in CRC patients. Specifically, high frequencies of CD4+ T cell subsets expressing PD-1 and CTLA-4 were associated with worse DFS in circulation (PMID: 35655158). However, high frequencies of Tregs/CD4+ T cells expressing TIM-3 in circulation were associated with longer DFS (PMID: 35655158). According to the reviewer’s comments, we have added new parts to the manuscript in the discussion and conclusion section. Page 10, lines 318-323 and 332-337, ref 42.

Reviewer 3 Report

In this paper, Mohammad et al. presented a study entitled "Correlations between circulating and tumor-infiltrating CD4+Treg subsets with immune checkpoints in colorectal cancer patients with early and advanced stages". Understanding the correlations between different immune checkpoints and Treg subsets in CRC patients may facilitate the understanding of Treg-mediated immunosuppression in cancer. The study is well-structured and the data presentation is straightfroward. Overall, I have some minor comments to the paper in current form.

1. In section 2.2, please specify the model of flow cytometer for FACS analysis.

2. In section 3.1, I suggest moving the description about Treg and Helios to Introduction. Only the result should be presented in this part.

Author Response

In this paper, Mohammad et al. presented a study entitled "Correlations between circulating and tumor-infiltrating CD4+Treg subsets with immune checkpoints in colorectal cancer patients with early and advanced stages". Understanding the correlations between different immune checkpoints and Treg subsets in CRC patients may facilitate the understanding of Treg-mediated immunosuppression in cancer. The study is well-structured and the data presentation is straightfroward. Overall, I have some minor comments to the paper in current form.

  1. In section 2.2, please specify the model of flow cytometer for FACS analysis.

Thank you for your suggestion. we have added the model of flow cytometer in section 2.2, page 2, lines 76-78.

  1. In section 3.1, I suggest moving the description about Treg and Helios to Introduction. Only the result should be presented in this part.

Thank you for your suggestion. We have moved the description about Treg and Helios to the introduction, page 2, lines 44-46 and 50-51.

Reviewer 4 Report

In this article, the authors perform a correlation analysis of Treg subpopulations with infiltrating CD4+ T cells expressing immune checkpoint inhibitors in CRC patients, based on datasets from a previous clinical study. The study that contains the experimental protocol and initial findings is: Toor SM, Murshed K, Al-Dhaheri M, Khawar M, Abu Nada M, Elkord E. Immune Checkpoints in Circulating and Tumor-Infiltrating CD4+ T Cell Subsets in Colorectal Cancer Patients. Front Immunol. 2019, 10:2936. doi: 10.3389/fimmu.2019.02936. The study cited in the text ([15]: Al-Mterin, M.A., K. Murshed, and E. Elkord, Correlations between Circulating and Tumor-Infiltrating CD4(+) T Cell Subsets with Immune Checkpoints in Colorectal Cancer. Vaccines (Basel), 2022. 10(4).) is a preceding correlational study using the datasets presented at Toor et al., and not the article that first presents the clinical findings. The present study presents major similarities with the two previous articles, and thus the novelty of the findings is questionable. More specifically, this study includes disease stage in the correlation analysis (which is the biggest difference with ref 15), however, staging was also examined in the initial clinical study (Toor et al., 2019).

Conclusively, although the issue of efficacious ICI treatment of CRC is very prominent, the lack of novelty of findings in the present study poses a major concern for its publication.

Author Response

There are different novel aspects in this study. As reviewer 1 correctly mentioned in their comments that “correlations between distinct Treg subsets and CD4+ T cells expressing ICs in CRC patients according to disease stages had not been investigated so far. Therefore, this study appears to be innovative and the data reported by the authors provide relevant indications on the disease mechanism in the whole organism and on prognosis”.

In our previous study in 2019 in Frontiers in Immunology, we ONLY investigated the expression levels of the immune checkpoints including PD-1, TIM-3, LAG-3, and CTLA-4 in different CD4+ T cell subsets, and we did not investigate any correlations at all between these immune checkpoints and different CD4+ T cell subsets. However, in this study, data presented are novel and different from our previous study. Also, we investigated the correlations of different FoxP3+/-Helios+/- T cell subsets with immune checkpoint-expressing CD4+ T cells in CRC patients with different disease stages. Specifically, in the TME, we found that the correlations between different FoxP3+/-Helios+/- T cell subsets with CD4+LAG-3+ T cells and CD4+CTLA-4+ T cells were higher in patients with early stages. However, the correlations between FoxP3+ Tregs, FoxP3+Helios+ Tregs, and FoxP3-Helios- T cells with CD4+TIM-3+ T cells were higher in patients with advanced stages. This is the first study to explore correlations of Treg subpopulations with immune checkpoint-expressing CD4+ T cells in CRC based on CRC staging.

Round 2

Reviewer 2 Report

No further comments.

Reviewer 4 Report

The authors answered my concerns satisfactorily.  Their study is suitable for publication.